# Breast Cancer Tumor Classification Using a Bag of Deep Multi-Resolution Convolutional Features

David Clement [1,2,3], Emmanuel Agu [2,*], John Obayemi [2], Steve Adeshina [3] and Wole Soboyejo [2]

1   Department of Computer Science, African University of Science and Technology, Abuja 900109, Nigeria
2   Worcester Polytechnic Institute, Worcester, MA 01609, USA
3   Computer Science Department, Nile University of Abuja, Abuja 900001, Nigeria
*   Correspondence: emmanuel@wpi.edu

**Abstract:** Breast cancer accounts for 30% of all female cancers. Accurately distinguishing dangerous malignant tumors from benign harmless ones is key to ensuring patients receive lifesaving treatments on time. However, as doctors currently do not identify 10% to 30% of breast cancers during regular assessment, automated methods to detect malignant tumors are desirable. Although several computerized methods for breast cancer classification have been proposed, convolutional neural networks (CNNs) have demonstrably outperformed other approaches. In this paper, we propose an automated method for the binary classification of breast cancer tumors as either malignant or benign that utilizes a bag of deep multi-resolution convolutional features (BoDMCF) extracted from histopathological images at four resolutions ($40\times$, $100\times$, $200\times$, and $400\times$) by three pre-trained state-of-the-art deep CNN models: ResNet-50, EfficientNetb0, and Inception-v3. The BoDMCF extracted by the pre-trained CNNs were pooled using global average pooling and classified using the support vector machine (SVM) classifier. While some prior work has utilized CNNs for breast cancer classification, they did not explore using CNNs to extract and pool a bag of deep multi-resolution features. Other prior work utilized CNNs for deep multi-resolution feature extraction from chest X-ray radiographs to detect other conditions such as pneumoconiosis but not for breast cancer detection from histopathological images. In rigorous evaluation experiments, our deep BoDMCF feature approach with global pooling achieved an average accuracy of 99.92%, sensitivity of 0.9987, specificity (or recall) of 0.9797, positive prediction value (PPV) or precision of 0.99870, F1-Score of 0.9987, MCC of 0.9980, Kappa of 0.8368, and AUC of 0.9990 on the publicly available BreaKHis breast cancer image dataset. The proposed approach outperforms the prior state of the art for histopathological breast cancer classification as well as a comprehensive set of CNN baselines, including ResNet18, InceptionV3, DenseNet201, EfficientNetb0, SqueezeNet, and ShuffleNet, when classifying images at any individual resolutions ($40\times$, $100\times$, $200\times$ or $400\times$) or when SVM is used to classify a BoDMCF extracted using any single pre-trained CNN model. We also demonstrate through a carefully constructed set of experiments that each component of our approach contributes non-trivially to its superior performance including transfer learning (pre-training and fine-tuning), deep feature extraction at multiple resolutions, global pooling of deep multiresolution features into a powerful BoDMCF representation, and classification using SVM.

**Keywords:** breast cancer; classification; deep convolutional neural networks; deep features; bag of convolutional features; malignant tumors; support vector machine (SVM)

## 1. Introduction

Breast cancer accounts for 30% of all female cancers [1,2], has the highest death rate of all types of cancers [1], and the number of new cases is expected to rise by almost 70% in the next two decades. There are two kinds of growth in breast tissue: non-harmful (benign) and dangerous (malignant or cancerous) that should be distinguished from each other during patient assessments. The World Health Organization (WHO) has stated that in

order to increase the survival rates of patients with breast cancer from 30% to 50%, early and precise diagnosis of malignancy is important [3]. However, due to variability in the availability and know-how of experts, 10% to 30% of breast cancers are not detected during regular assessment [4].

Computer-assisted diagnosis systems (CAD) for breast cancer detection have been proposed to automate recognizing cancerous regions, for distinguishing normal vs. abnormal tissues (tumors), and malignant vs benign tumors, increasing accuracy by up to 10% [5]. CAD systems are fast, readily accessible, and dependable for early diagnoses [6]. In most contemporary CAD frameworks, machine learning is adopted for medical image analysis, breast cancer detection, and diagnosis [3]. Automated therapeutic imaging techniques including breast X-ray images, sonograms methods, magnetic resonance imaging, computed tomography, and histopathological imaging are compelling for breast cancer detection [7,8], as the accuracy of manual breast cancer screening varies depending on the pathologist's experience and knowledge. Human errors can occur, resulting in incorrect diagnoses. Histopathological images are currently considered the highest quality for the clinical identification of cancer [9]. Automated and exact classification of histopathological images is the foundation of many top-down and bottom-up image analyses such as nuclei classification, mitosis detection, and gland segmentation [10]. However, of all histopathological image examination tasks, tumor classification is the most important. Earlier image-based breast cancer classification research utilized machine learning (ML) with handcrafted image features [11–14]. However, due to their impressive performance in computer vision and image processing tasks, approaches utilizing neural networks have recently become popular. Convolutional neural networks (CNNs) have demonstrated superior performance for a wide range of image analyses tasks, including image classification, ailment detection, localization, segmentation [15], and the analyses of histopathological images [16].

Our approach: In this paper, we propose an ML method for binary classification (malignant vs. benign) of breast histopathological breast cancer images. First, deep multi-resolution convolutional features are extracted from four resolutions ($40\times$, $100\times$, $200\times$, and $400\times$) of histopathological breast cancer images using three state-of-the-art CNN-based backbone models: (1) (Efficientnet-b0) [17], (2) Inception deep architecture (Inception-v3) [18], and (3) ResNet50 [19]. The multiresolution CNN features are then pooled using global average pooling to create a bag of deep multiresolution convolutional features (BoDMCF), which is then classified using a support vector machine (SVM) classifier [20]. Inception-V3 permits deeper neural networks without increasing parameters and contains Inception modules, which achieve dimensionality reduction with stacked $1 \times 1$ convolutions. EfficientNetb0 is the baseline model for EfficientNet, which utilizes compound scaling, a novel scaling method, to scale the dimensions of the model uniformly, resulting in increased performance. The ResNet-50 model of deep residual networks is a CNN with 50 layers, which stacks residual blocks, mitigating the vanishing gradient descent problem in order to maintain accuracy as the model becomes deeper. In medical image analysis using ML, feature extraction is a fundamental image analysis step, and various extraction strategies have been proposed for image-based classification of various ailments in prior work [21–25]. There are three main classes of image feature extraction methods [26]: (1) handcrafted features, (2) unsupervised feature learning, and (3) deep feature learning. Handcrafted feature extraction is tedious and error prone. In this paper, feature and representation auto-learning using pre-trained, state-of-the-art deep CNN models is utilized.

In rigorous evaluation experiments, our deep BoDMCF feature approach with global pooling achieved an average accuracy of 99.92%, sensitivity of 0.9987, specificity (or recall) of 0.9797, positive prediction value (PPV) or precision of 0.99870, F1-Score of 0.9987, MCC of 0.9980, Kappa of 0.8368, and AUC of 0.9990 on the BreaKHis dataset [27]. The deep BoDMCF approach outperforms the prior state of the art for classifying histopathological breast cancer images and a comprehensive set of state-of-the-art CNN baselines including ResNet18, InceptionV3, DenseNet201, EfficientNetb0, SqueezeNet, and ShuffleNet

when classifying any single resolution ($40\times$, $100\times$, $200\times$ or $400\times$). In our evaluation, we demonstrate through a carefully constructed set of experiments that each component of our approach contributes non-trivially to its superior performance, including transfer learning pre-training and fine-tuning, deep feature extraction at multiple resolutions, global pooling of deep multiresolution features into a powerful BoDMCF representation, and classification using SVM.

Novelty: Our work is novel because while some prior work has utilized CNNs for breast cancer classification, they did not explore using CNNs to extract and pool a bag of deep multi-resolution features. Other prior work utilized CNNs for deep multi-resolution feature extraction from chest X-ray radiographs to detect other conditions such as pneumoconiosis but not for breast cancer detection from histopathological images. The BoDMCF approach innovatively leverages several key insights. First, pre-training state-of-the-art CNNs on larger repositories such as the 14 million image ImageNet repository equips them with the intelligence to learn the most predictive features and low-level image attributes such as edges and corners from histopathological breast cancer images. Secondly, extracting and pooling features from multiple resolutions of histopathological images improves classification accuracy as discriminative visual attributes may be most visible at different resolutions. Thirdly, global pooling of multiresolution breast cancer features creates a bag of features that is so powerful that classifying them using SVM achieves highly accurate binary breast cancer classification (malignant vs. benign) of histopathological images. The deep BoDMCF approach has yielded impressive results in other image classification domains including multimedia image retrieval [28] and remote sensing image scene classification [26]. Ours is the first work to innovatively apply this powerful representation learning technique to binary breast cancer image classification (malignant vs. benign). The specific combination of state-of-the-art deep learning architectures we utilize are also novel and were carefully selected after extensive, systematic experimentation.

Challenges: First, the heterogeneity of the visual texture patterns observable on breast histopathological images makes tumor malignancy classification a challenging task even for CNNs, affecting their performance [29]. Secondly, the discriminative visual attributes of tumor malignancy can be most visible at different resolutions of histopathological images. By directly addressing these two challenges, the BoDMCF approach is particularly suited to classifying tumor malignancy.

Related work that utilized Deep Learning and CNNs for breast cancer tumor classification are summarized in Table 1. While there has been some prior work that utilized neural networks for breast cancer classification, none of them explored the deep BoDMCF representation with a global pooling approach, which we propose. Maqsood et al. in [30] classified screening mammogram using CNN and achieved average accuracy of 97.49%. Spanhol et al. [31] utilized the AlexNet CNN model for classifying tumors in histopathological images as malignant or benign. Kowal et al. [32] explored deep learning models for nuclei segmentation, in which the instances were classified as harmless or non-harmless on a dataset of 269 images, achieving average accuracies from 80.2% to 92.4%. Shen et al. [33] utilized an active learning approach to classify breast cancer images. Byra et al. [34] combined statistical parameters with a CNN for breast cancer classification. Nejad et al. [35] used a fast one-layer CNN for breast cancer classification that was tested on histopathological images with a magnification factor of $40\times$. Nahid et al. in [36] used DNN models guided by unsupervised clustering methods for breast cancer classification. Murtaza et al. [3] comprehensively reviewed cutting-edge deep-learning-based breast cancer classification using medical images. Ogundokun et al. [37] utilized artificial neural networks and CNNs with hyperparameter optimization for malignant vs. benign classification, while the support vector machine (SVM) and multilayer perceptron (MLP) were utilized as baseline classifiers for comparison. Vogado et al. [38] proposed a technique used to correctly classify images with different characteristics derived from different image databases which does not require a segmentation process. Gandomkar et al. [39] classified breast histopathological images into malignant and benign subtypes using deep residual networks. Han et al. [40]

previously utilized deep neural networks in classifying histopathological breast images into their sub-types and used majority voting for patient classification. Whilst their work focused on classifying different breast histopathological images into their sub-types and achieved 93.2% accuracy, we performed binary classification using BoDMCF extracted from breast histopathological images without considering their image subtypes.

**Table 1.** Related Work on deep learning and CNNs for breast cancer tumor classification.

| Authors | Models | ML Problem | Summary of Approach | Accuracy |
|---|---|---|---|---|
| R. Yan et al. [10] | CNN and RNN | Four-class classification into malignant and benign subtypes | A CNN was used to extract image patches. Then an RNN was used to fuse the patch features and make the final image classification. | 91.3% |
| M. Amrane [14] | Naive Bayes (NB) and k-nearest neighbor (KNN) | Binary classification (malignant or benign) | For NB, data was split into blocks of 2 classes and 2 sets of features T and classes D and statistical analysis were performed. K-nearest neighbor, pick an instance from the testing sets and calculate its distance with the training set. | 97.51% for KNN and 96.19% for NB |
| S. H. Kassani, M. J. Wesolowski, and K. A. Schneider [29] | VGG19, MobileNet, and DenseNet. | Binary classification (malignant or benign) | Ensemble model was used for the feature extraction. Then classification was done using a Multi-Layer Perceptron (MLP) classifier | 98.13% |
| F. A. Spanhol, L. S. Oliveira, C. Petitjean, and L. Heutte [31] | Ensemble models | Binary classification (malignant or benign) | Various CNNs were using a fusion rule for breast cancer classification | 85.6% |
| Kowal et al. [32] | Deep learning model | Binary classification (malignant or benign) | Segmentation, feature extraction and classification were performed on individual cell nuclei of cytological images. | 92.4% |
| A. Al Nahid, M. A. Mehrabi, and Y. Kong [36] | CNN, LSTM, K-means clustering, Mean-Shift clustering and SVM | Binary classification (malignant or benign) | A set of biomedical breast cancer images were classified using novel DNN models guided by an unsupervised clustering method | 96.0% |
| Z. Gandomkar, P. C. Brennan, and C. Mello-Thoms [39] | Deep residual network (ResNet) | Multi class classification into Subtypes of malignant and benign | Approach consisted of two stages. In the first stage, ResNet layers classified patches from the images as benign or malignant. In the second statge, images were classified into subtypes of malignant and benign | 98.52%, 97.90%, 98.33%, and 97.66% in 40×, 100×, 200× and 400× magnification factors respectively |

Related work that used CNNs to extract deep features from medical images Wichakam et al. proposed an automated system that uses a CNN for feature extraction and an SVM for classification for mass detection on digital mammographic images [41] but did not explore multi-resolution extraction and pooling to create a bag of deep features. Devnath et al. [42] used CNN models for automated detection of pneumoconiosis by extracting deep multi-level features from X-ray images that were then classified using SVM. Devnath et al. [43] conducted a systematic review of computer-aided diagnosis of coal workers' pneumoconiosis in chest X-ray radiographs using machine learning, which included approaches that utilized CNNs for feature extraction. Devnath et al. [44] utilized the CheXNet-121 model as a feature extractor as part of a method for detecting and visualizing pneumoconiosis using an ensemble of multi-dimensional deep features learned from chest X-rays. Firstly, they removed the last layer close to the output layer; next, a global average pooling layer was added which converted the output of the model into one-dimensional vectors. Huynh et al. [45] tested the optimal point at which to extract features from pre-trained CNN, identifying the specific utility of transfer learning in computer-aided diagnosis (CADx) systems. Zhang et al. [46] proposed to build ensemble learners through fusing multiple deep CNN learners for pulmonary nodule classification. Other related work includes research by Filipczuk et al. [47] and George et al., who previously extracted nuclei features from fine needle biopsies. First, the circular Hough transform was utilized

for detecting nuclei candidates and false-positive reduction, followed by using machine learning and Otsu thresholding [48].

The rest of this paper is structured as follows. Section 2 presents some background required to understand our work, including introducing the BreaKHis database, basic concepts of CNNs, and a description of the pre-trained CNNs we explored for feature extraction. Our proposed BoDMCF representation and machine learning methodology is presented in Section 3. Section 4 presents our experimental results, and Section 5 discusses our findings and Section 6 concludes the paper.

## 2. Background

### 2.1. BreakHis Breast Cancer Histopathological Image Dataset

Our neural networks breast cancer models were created by analyzing the BreaKHis database [27], which contains 7,909 tiny histopathological biopsy images of benign and malignant breast tumors. The distribution of images in the BreakHis database is summarized in Table 2. In an IRB-approved study, patients with traces of breast cancer who visited the the P&D, Brazil, between January to December 2014 were recruited. Those who agreed to participate properly consented. Breast tissue biopsy test slides were created by staining the samples with hematoxylin and eosin, prepared for histopathological examination, and marked by pathologists at the P&D Lab. The widely accepted paraffin preparation methodology was utilized. The overall preparation technique incorporates several steps including fixation, dehydration, clearing, infiltration, inserting, and cutting [49]. Lastly, an experienced pathologist diagnosed every case, which was confirmed by correlative tests, such as by utilizing the immunohistochemistry assessment. An Olympus BX-50 magnifying device having a transfer focal point and magnification of $3.3\times$ fixed to a Samsung sophisticated digital camera SCC-131AN was employed to acquire digitized pictures from the breast tissue slides. Images were procured in red, green, and blue channels (RGB) color space (3-byte color depth, 1 byte for every color channel) utilizing amplifying factors of $40\times$, $100\times$, $200\times$, and $400\times$ and comparing the variables to target main points of $4\times$, $10\times$, $20\times$, and $40\times$. Four images—at the four amplification factors: (a) $40\times$, (b) $100\times$, (c) $200\times$, and (d) $400\times$ were generated. Images generated from a single slide of breast tissue containing a malignant tumor (breast cancer) are shown in Figure 1.

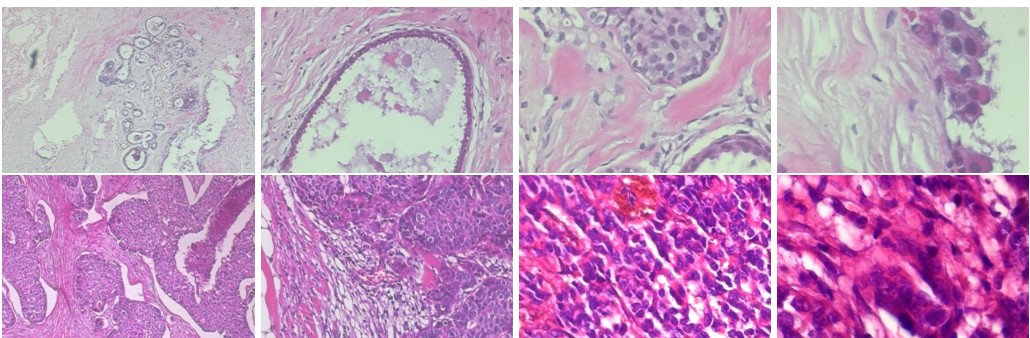

**Figure 1.** Sample histopathological images. The **top row** shows benign images at $40\times$, $100\times$, $200\times$, and $400\times$ (left to right). The **bottom row** shows malignant images at $40\times$, $100\times$, $200\times$, and $400\times$ (left to right).

**Table 2.** Distribution of the histopathological breast cancer images in the BreakHis dataset by amplication factor and class.

| Magnification | Malignant | Benign | Total |
|---|---|---|---|
| 40× | 1370 (68.67%) | 652 (32.68%) | 1995 |
| 100× | 1437 (69.05%) | 644 (30.95%) | 2081 |
| 200× | 1390 (69.05%) | 623 (30.94%) | 2013 |
| 400× | 1232 (67.69%) | 588 (32.31%) | 1820 |
| Total | 5429 (68.64%) | 2480 (31.14%) | 7909 |

*2.2. Convolutional Neural Networks (CNNs)*

Convolutional neural networks (CNNs) have recently become the best performing neural networks for image analyses and classification. The BoDMCF approach utilizes pre-trained, state-of-the-art CNN models for feature extraction. This section provides a summary of some of the technical details of the CNN architecture. CNNs are in the category of feedforward neural network (FFN) models, where the signal passes within the network without a loop back and can be expressed in Equation (1) [50].

$$G(x) = g_H(g_{H-1}(\ldots(g_1(x))))$$ (1)

where $H$ indicates the number of hidden layers, and $g_i$ denotes the function in the matching layer $i$. The core functional layers in a typical CNN model incorporate activation, fully connected (FC), pooling layers, and a classification layer. The convolutional layer, $f$, is comprised of various convolutional kernels $(f_1 \ldots f_{y-1}, f_y)$ where every $f_y$ denotes a linear function in the $y^{th}$ kernel that can be represented by Equation (2)

$$f^y(x,j) = \sum_{u=-m}^{m} \sum_{v=-n}^{n} \sum_{d=-w}^{w} w_y(u,v,w) I(x-u, j-v, z-w)$$ (2)

The position of the pixel in the input $I$ is denoted by the coordinates $(x, j, z)$, the weight for the $y^{th}$ kernel is denoted by $W_y$, and the height, width, and depth of the filter is denoted by $m$, $n$, and $w$. The rectified linear unit (ReLU) is a pixel-wise non-linear function, $g$, known as the activation layer, is represented in Equation (3) [50–52].

$$g(x) = \max(0, x)$$ (3)

The pooling layer, $k$, is a layered non-linear down-sampling function designed to repeatedly decrease the feature representation size. The FC layer is considered a variation of the convolutional layer whose kernel has the size $1 \times 1$. The classification SoftMax layer ($\sigma(\vec{z})_i = \frac{e^{z_i}}{\sum_{j=1}^{K} e^{z_j}}$) is typically added to the last fully connected layer to calculate the probabilities of $I_i$ fit into different classes. Figure 2 shows a simple example of a CNN model that is made up of convolutional, ReLU, max-pooling, and FC layers. The first, second, and fifth ReLu layers precede the maximum-pooling layer, which in turn precedes the three FC layers. In order to express max-pooling formally, let $\mathbf{Z}'$ be a $n_l \times n_l \times m_l$ tensor. Max-pooling involves determining the maximum value over the element-wise product of subtensor $\mathbf{Z}_k^l(i, j, q)$ and filter $\mathbf{W}$, given by Equation (4).

$$\max\left(\mathbf{Z}_k^l(i,j,q) \odot \mathbf{W}\right) = \max_{\substack{a=1,2,\cdots,k \\ b=1,2,\cdots,k \\ c=1,2,\cdots,r}} \left\{ z_{i+a-1, j+b-1, q+c-1}^l \cdot w_{a,b,c} \right\}$$ (4)

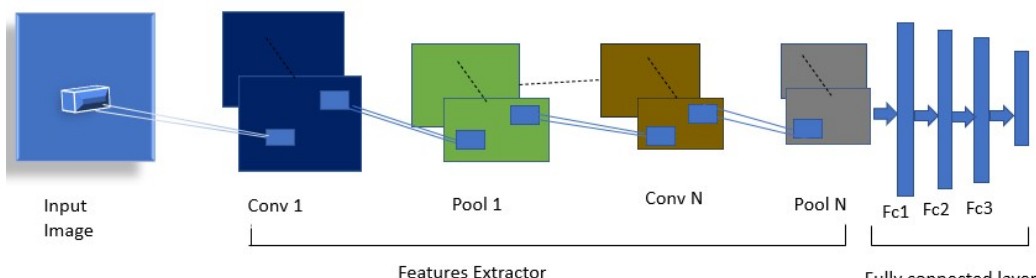

**Figure 2.** The architecture of a convolutional neural network (CNN).

### 2.3. Pre-Trained CNNs for Deep Image Feature Extraction

To create the bag of deep multi-resolution convolutional features (BoDMCF) representation, features are extracted from four resolutions ($40\times$, $100\times$, $200\times$, and $400\times$) of histopathological breast cancer images using three (3) state-of-the-art CNN-based models: (1) (Efficientnet-b0) [17], (2) Inception deep CNN architecture (Inception-v3) [18], (3) ResNet50 [19]. These models were pre-trained on the ImageNet repository that has 14 million images in 1000 categories, enabling them to gain significant intelligence about images [53]. Pre-training is part of a transfer learning approach, which yields higher starting/initial model accuracy during training, faster convergence; and higher asymptotic accuracy (the accuracy level to which the training converges). We now provide some background on these state-of-the-art deep CNN image classification models.

EfficientNet [17]: This architecture and scaling method utilizes a compound coefficient to uniformly scale all depth, width, and resolution dimensions of the CNN using a set of fixed scale coefficients. Given a ConvNet defined as $\mathcal{N} = \odot_{i=1...s} \mathcal{F}_i^{L_i}\left(X_{\langle H_i, W_i, C_i \rangle}\right)$, the EfficientNet architecture can be formulated as an optimization problem given by Equation (5)

$$
\begin{aligned}
\max_{d,w,r} \; & \text{Accuracy}(\mathcal{N}(d,w,r)) \\
\text{s.t. } & \mathcal{N}(d,w,r) = \underset{i=1...s}{\odot} \hat{\mathcal{F}}_i^{d \cdot \hat{L}_i}\left(X_{\langle r \cdot \hat{H}_i, r \cdot \hat{W}_i, w \cdot \hat{C}_i \rangle}\right) \\
& \text{Memory}(\mathcal{N}) \leq \text{target\_memory} \\
& \text{FLOPS}(\mathcal{N}) \leq \text{target\_flops}
\end{aligned}
\tag{5}
$$

In a principled manner, EfficientNet scales network width, depth, and resolution based on a single $\delta$ compound coefficient as expressed in Equation (6).

$$
\begin{aligned}
\text{depth: } & d = \alpha^\phi \\
\text{width: } & w = \beta^\phi \\
\text{resolution: } & r = \gamma^\phi \\
\text{s.t. } & \alpha \cdot \beta^2 \cdot \gamma^2 \approx 2 \\
& \alpha \geq 1, \beta \geq 1, \gamma \geq 1
\end{aligned}
\tag{6}
$$

For instance, in order to utilize 2N times more computational resources, the network depth can simply be increased by $\alpha N$, the width by $\beta N$, and the image size by $\gamma N$, where $\alpha$, $\beta$, and $\gamma$ are constant coefficients determined by a small grid search on the original small model. In order to capture more fine-grained patterns from a larger input image, the compound scaling method uses more layers to increase the receptive field and more channels to capture a larger number of channels. MobileNet-V2's [49] inverted bottleneck residual blocks along with squeeze-and-excite blocks are the basis of EfficientNet-B0's base network. Figure 3 is the architecture for the EfficientNet B0 model.

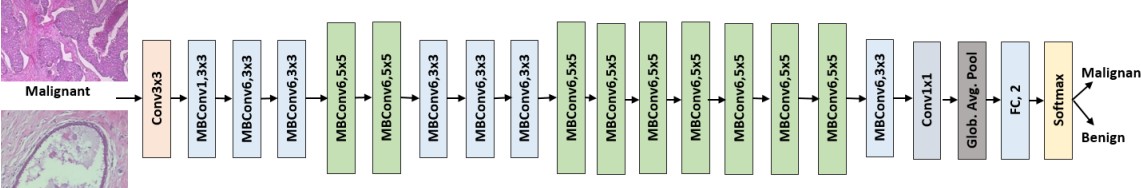

**Figure 3.** EfficientNet B0 CNN architecture.

Inception: This architecture has introduced multiple versions. The first version of the Inception CNN model was introduced as GoogLeNet [54], named Inceptionv1. The enhanced usage of computing resources within the inception1 network is the fundamental feature of this architecture, accomplished by increasing the network's depth and depth, while sustaining the computational budget. Version 2, also named (Inception-v2), incorporated batch normalization [55]. Version 3 (called Inception-v3) utilized additional factorization ideas [18] . The main distinction of Inception-V3 is that $5 \times 5$ convolutional layers were used instead of two consecutive layers of $3 \times 3$ convolutions with up to 128 filters and also the addition of a Batch Norm (BN)-auxiliary. A BN auxiliary is a version of the auxiliary classifier in which the fully connected layer is also normalized in addition to the convolutions. The RMSProp optimizer was also utilized, which has an update rule that can be expressed as:

$$E\left[g^2\right]_t = \beta E\left[g^2\right]_{t-1} + (1 - \beta)\left(\frac{\delta C}{\delta w}\right)^2$$
$$w_t = w_{t-1} - \frac{\eta}{\sqrt{E[g^2]_t}}\frac{\delta C}{\delta w} \tag{7}$$

where $E(g)$ is the moving average of squared gradients, $(\frac{\delta C}{\delta w})^2$ is the gradient of the cost function with respect to the weight, $\eta$ is the learning rate, and $\beta$ is the moving average parameter. The classification layers utilized label smoothing regularization (LSR). LSR can be obtained by replacing a single cross entropy $H(q, p)$ in the loss function with a pair of losses in the cross entropy, $H(q, p)$ and $H(u, p)$, as given in Equation (8) below. The second loss penalizes the deviation of predicted label distribution $p$ from the prior $u$, with the relative weight $\frac{\epsilon}{1-\epsilon}$. $H(u, p)$ is a measure of how dissimilar the predicted distribution $p$ is to uniform.

$$H(q', p) = -\sum_{k=1}^{K} \log p(k)q'(k) = (1 - \epsilon)H(q, p) + \epsilon H(u, p) \tag{8}$$

The model is 48 layers deep and capable of classifying images into 1000 image classes, including various object types, keyboard, mouse, pencil, and different animals. This pretraining ensures that the model has gained knowledge of deep high-level feature depictions of an extensive variety of images. Figure 4 is the architecture for the Inception-v3 model.

ResNet: This architecture introduced a deep residual learning structure, which reformulates the CNN's layers as learning residual functions of the layer inputs. Correctly denoting the desired underlying mapping as $K(i)$, the stacked non-linear layers were made to fit another mapping of $E(i) := K(i) - i$. ResNet solved the vanishing gradient, whereby the value of the neural network's gradient decreases significantly during backpropagation until its weights barely change. ResNet solved the vanishing gradient problem using a skip connection by adding the original input to the output of the convolutional block. A skip connection is a direct connection that skips over some of the model layers and can be

expressed as $\mathbf{y} = \mathcal{F}(\mathbf{x}, \{W_i\}) + W_s\mathbf{x}$, where $\mathcal{F}(\mathbf{x}, \{W_i\})$ represents the residual mapping to be learned. Resnet utilizes the SGD optimizer with momentum given by Equation (9)

$$v_t = \rho v_{t-1} + \nabla f(x_{t-1}) x_t = x_{t-1} - \alpha v_t \tag{9}$$

where $v_{t+1}$ is the momentum value, $\rho$ is a friction, $\nabla f(x_{t-1})$ is the gradient of the objective function at iteration $t-1$, $x_t$ are parameters, and $\alpha$ is the learning rate. ResNet50 [19], which our approach utilized, is a variant of ResNet. It has 48 convolutional layers and 1 MaxPool layer as well as an average pool layer. Figure 5 is the architecture for the ResNet model.

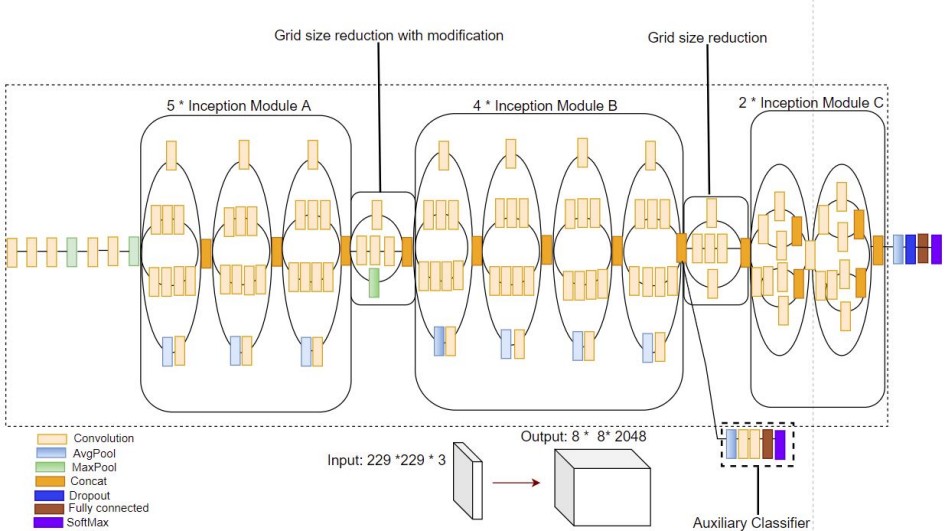

**Figure 4.** Inception-V3 CNN architecture.

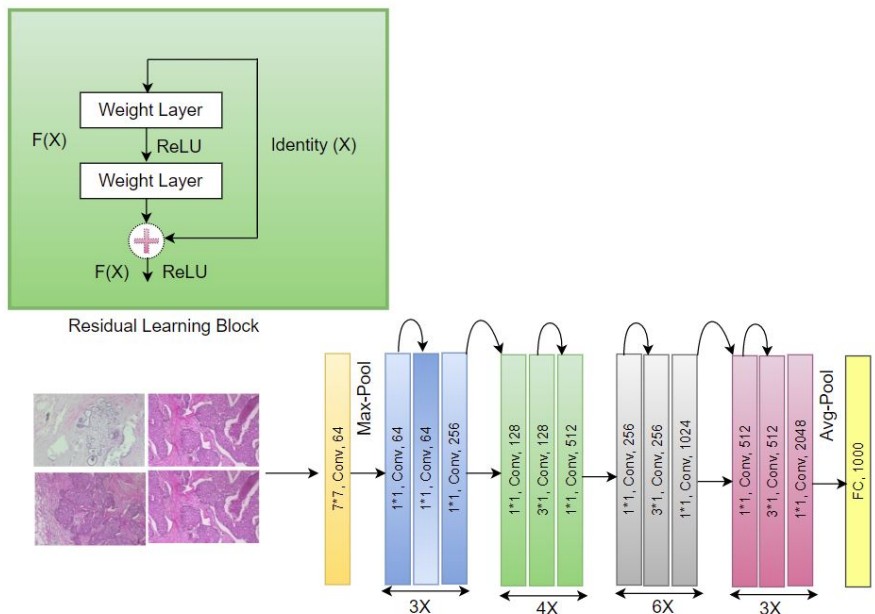

**Figure 5.** Resnet-50 CNN architecture.

## 3. Materials and Methods

Our overall approach involves extracting deep multi-resolution features from four resolutions ($40\times$, $100\times$, $200\times$, and $400\times$) of high resolution ($2048 \times 1536$) histopathological breast cancer images using the Efficientnet-b0 [17], Inception-v3 [18], and ResNet50 [19] pre-trained image pre-trained CNN models that are pooled using global pooling to create a BoDMCF. A support vector machine (SVM) classifier then uses the BoDMCF to clas-

sify histopathological breast cancer images as either malignant or benign. As shown in Figure 6, the proposed framework of breast cancer classification consists of three main modules: (i) data pre-processin,g (ii) deep BoDMCF feature extraction, and (iii) classification using SVM.

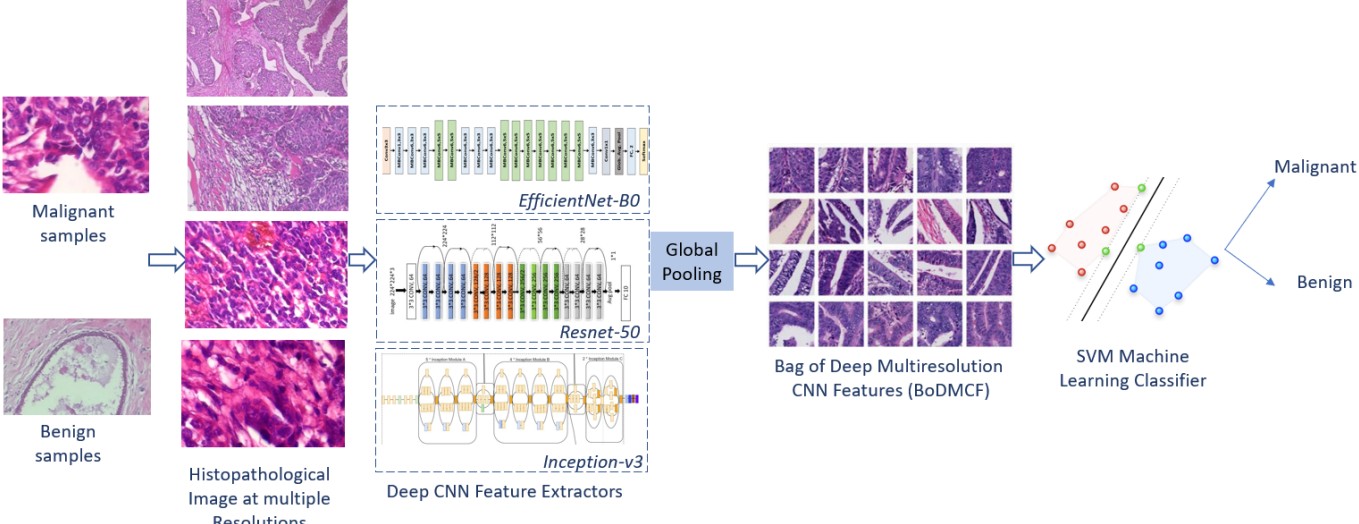

**Figure 6.** Our proposed approach: A bag of multiresolution CNN features (BoDMCF) are extracted from multiple resolutions of samples of malignant and benign histopathological images, which are then classified using a support vector machine (SVM) classifier.

*3.1. Step 1: Histopathological Image Pre-Processing*

During this step, each histopathological image is resized to fit into an input size suitable for different deep CNN models. The histopathological images were resized from $2048 \times 1536$ to $299 \times 299$ for inception-v3 and EfficientNet-B0 and to $224 \times 224$ for resnet-50. Random color data augmentation was also performed on each image by changing the brightness of the image randomly between 50% $(1 - 0.5)$ and 150% $(1 + 0.5)$ of the original image. (See Figure 7) Data augmentation generates diverse samples, which enables the model to learn a robust representation that is invariant to minor changes [56]. Examples of resized histopathological images are shown in Figure 8. After pre-processing, training and test sets were created using a 70:30 split ratio.

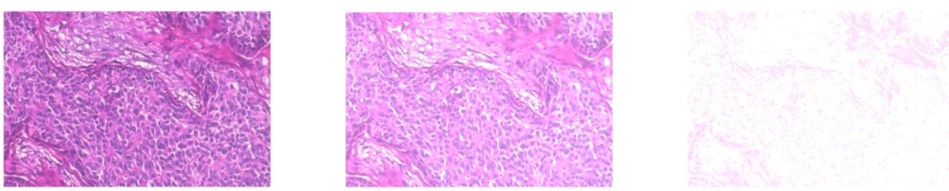

**Figure 7.** Example of brightness data augmentation operations (left = original, middle = brightness + 50, right = brightness + 150.

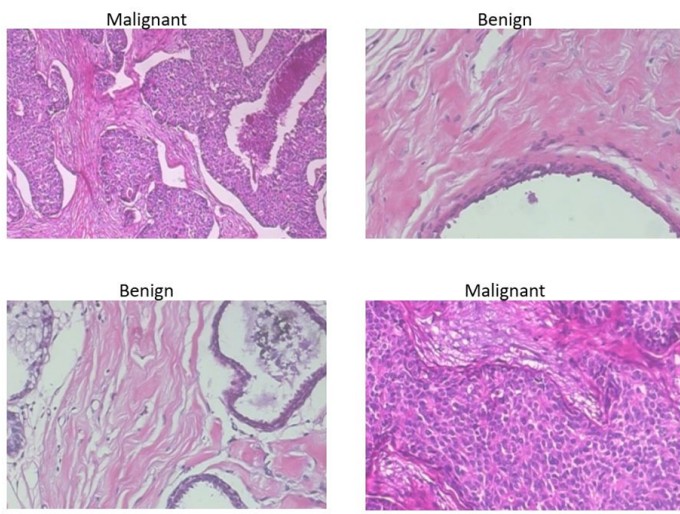

**Figure 8.** Examples of images after resizing to dimensions 224 × 224 × 3.

### 3.2. Step 2: Deep Multi-Resolution Feature Extraction Using Pre-Trained CNNs

This stage involves extracting the BoDMCF by modifying the final layers of the three pre-trained deep convolutional networks: Efficientnet-b0, Inception-v3, and ResNet50. These pre-trained models were trained on full-sized ImageNet images, then transfer learning (fine-tuning) was performed on histopathological breast cancer images in our dataset. These feature extractor CNN models utilized layer activations as features. The rich multi-level activations (features) extracted from four resolutions of histopathological images were then pooled to form the BoDMCF and finally used to train a support vector machine (SVM).

EfficientNet [17]: The input size of Efficientnet-b0 was 224 × 224, and Table 3 shows the activation strengths of 56 features learned by the average pooling layer by setting channels to be the vector of indices 1:56 and setting pyramid levels to 3 (three) so that the images are not scaled.

**Table 3.** Activation strength on 56 features learned by the average pooling layer for efficientNet.

| Iteration | Activation Stength | Pyramid Level |
|:---:|:---:|:---:|
| 1 | 0.35 | 1 |
| 2 | 0.31 | 1 |
| 3 | 0.59 | 1 |
| 4 | 1.19 | 1 |
| 5 | 1.87 | 1 |
| 6 | 2.56 | 1 |
| 7 | 3.12 | 1 |
| 8 | 3.56 | 1 |
| 9 | 3.87 | 1 |
| 10 | 4.15 | 1 |

Inception-v3 [18]: The model accepts an image input size of 299 × 299. Table 4 shows the activation strength of 56 features learned by the average pooling layer by setting channels to be the vector of indices 1:56 and setting pyramid levels to 1 (one) so that images are not scaled.

**Table 4.** Activation strength of 56 features learned by the average pooling layer for Inception-V3.

| Iteration | Activation Strength | Pyramid Level |
|:---:|:---:|:---:|
| 1 | 0.32 | 1 |
| 2 | 0.35 | 1 |
| 3 | 0.58 | 1 |
| 4 | 0.98 | 1 |
| 5 | 1.51 | 1 |
| 6 | 1.94 | 1 |
| 7 | 2.37 | 1 |
| 8 | 2.73 | 1 |
| 9 | 3.09 | 1 |
| 10 | 3.29 | 1 |

ResNet-50 [19] The input size of ResNet18 is $224 \times 224$ and Table 5 shows the activation strengths of 56 features learned by the average pooling layer, derived by setting channels to be the vector of indices 1:56 and setting pyramid levels to 1 (one) so that the images are not scaled.

**Table 5.** Activation strength on 56 features learned by the average pooling layer for ResNet-18.

| Iteration | Activation Strength | Pyramid Level |
|:---:|:---:|:---:|
| 1 | 0.94 | 1 |
| 2 | 1.18 | 1 |
| 3 | 2.92 | 1 |
| 4 | 5.34 | 1 |
| 5 | 7.22 | 1 |
| 6 | 8.50 | 1 |
| 7 | 9.41 | 1 |
| 8 | 10.04 | 1 |
| 9 | 10.60 | 1 |
| 10 | 10.88 | 1 |

*3.3. Step 3: Global Pooling of Features to Create BoDMCF*

Features extracted by the three state-of-the-art CNN models (ResNet-50, InceptionV3, and Efficientnet-b0) were pooled to acquire high-quality image descriptions using the activations of the global pooling layers at the end of the network as shown in Figure 9. The network constructs a hierarchical representation of input images. Deeper layers contain higher-level features, constructed using the lower-level features of earlier layers. To obtain the feature representations of the training and test images, activations on the global pooling layer, 'avg_pool', at the end of the network are utilized. The global pooling layer pools the input features over all spatial locations, giving 512 features in total as described in Figure 9. For each spatial location, the $f$ activations maps labelled $f_1$, $f_2$, $f_3$, to $f_{512}$ are collected, forming $1 \times 1 \times f$ column features of dimensions 1,1, 1,2 to $h,w$. These multiple features are then concatenated into a BoDMCF that is classified using SVM.

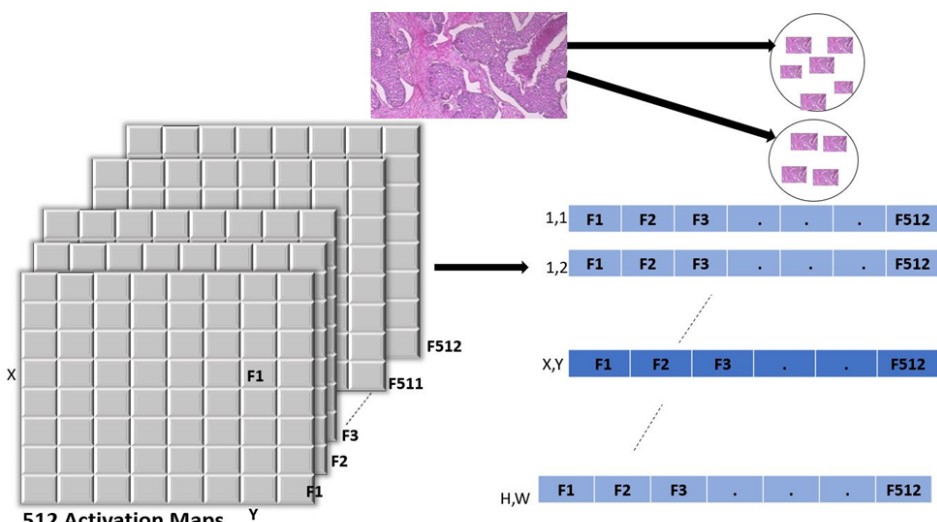

**Figure 9.** Feature pooling approach to create bag of deep multiresolution convolutional features (BoDMCF).

*3.4. Step 4: BoDMCF Classification Using SVM*

SVM was utilized to classify the BoDMCF extracted by the three CNN models as described above. Given a training set and class label $(B_n, A_n), n = 1, \ldots, N, B_n \in RD, A_n \in 1, 1$, the support vector machine (SVM) classifier [57] tries to find a hyperplane in feature space, which maximizes the margin between two classes (malignant vs. benign). SVM is based on the theory of maximum linear discriminants. For two classes to be classified, SVM finds peripheral data points in each class that are closest to the other class (called support vectors). For a dataset $D$ with $n$ points $x_i$ in a $d$-dimensional space, a hyperplane function $h(x)$ can be defined as

$$h(\mathbf{x}) = w^T \mathbf{x} + b = w_1 x_1 + w_2 x_2 + \ldots + w_d x_d + b \tag{10}$$

Overall, with $n$ points, the margin of the linear classifier can be defined as the minimum distance of a point from the separating hyperplane given as:

$$\delta^* = \min_{\mathbf{x}_i} \left\{ \frac{y_i \left( \mathbf{w}^T \mathbf{x}_i + b \right)}{\|\mathbf{w}\|} \right\} \tag{11}$$

The SVM classifier finds the optimal hyperplane dividing the two classes by solving the minimization problem with an objective function:

$$\min_{\mathbf{w}_i b} \left\{ \frac{\|\mathbf{w}\|^2}{2} \right\} \tag{12}$$

with linear constraints:

$$h(\mathbf{x}) = y_i \left( w^T \mathbf{x} + b \right) \geq 1, \forall \mathbf{x}_i \in \mathbf{D} \tag{13}$$

Then, the class of a new point is predicted as:

$$\hat{y} = \text{sign}(h(\mathbf{z})) = \text{sign} \left( w^T \mathbf{z} + b \right) \tag{14}$$

## 4. Evaluation and Results

*4.1. Evaluation Metrics*

The following metrics were used to evaluate all neural networks breast cancer classification models.

Accuracy (Acc): This demonstrates how many malignant cases are correctly predicted and how many benign cases are correctly diagnosed. Equation (15) describes it.

$$ACC = \frac{(TP + TN)}{\text{TP} + \text{TN} + \text{FP} + \text{FN}} \tag{15}$$

Sensitivity (Sens): This is the percentage of positive instances correctly predicted, which can be computed using Equation (16).

$$\text{Sens} = \frac{TP}{\text{TP} + \text{FN}} \tag{16}$$

Precision (Prec): This expresses how many of the positive predictions are actually correct as expressed as Equation (17).

$$\text{Prec} = \frac{TP}{\text{TP} + \text{FP}} \tag{17}$$

Specificity (Spec): This measures the percentage of correct negative predictions and can be expressed as Equation (18):

$$\text{Spec} = \frac{TN}{\text{TN} + \text{FP}} \tag{18}$$

F1-score (Fscore): This analyzes sensitivity and precision in harmony by applying a penalty to extreme values in order to reflect their simultaneous impact and can be expressed as Equation (19).

$$\text{Fscore} = \frac{TP}{\text{TP} + \frac{1}{2}(\text{FP} + \text{FN})} \tag{19}$$

AUC: This is a probability curve that plots the True Postive Rate (TPR) against the False Positive Rate (FPR) at various threshold values and essentially separates the 'signal' from the 'noise' and is expressed as Equation (20). AUC is a number that ranges from 0 to 1. An AUC value of one indicates a perfect model, while an AUC of 0.5 or below indicates an inadequate model.

$$AUC = \frac{\sum_i R_i(I_p) - I_p(I_p + 1)/2}{I_p + I_n} \tag{20}$$

where $I_p$ and $I_n$ denote the number of malignant and benign breast images, respectively, and $R_i$ is the rank of the $i$th positive image in the ranked list.

The Matthews Correlation Coefficient (MCC): This is a contingency matrix metric for calculating the Pearson product-moment correlation coefficient between actual and predicted values that is unaffected by the unbalanced datasets issue. MCC can be expressed as Equation (21).

$$MCC = \frac{TP \cdot TN - FP \cdot FN}{\sqrt{(TP + FP) \cdot (TP + FN) \cdot (TN + FP) \cdot (TN + FN)}} \tag{21}$$

Kappa (Kapp): This is a statistic that compares observed and expected accuracy. It is a measure of how well the instances categorized by a classifier matched the data designated as ground truth. Equation (22) can be used to calculate Kappa.

$$\text{Kappa} = \frac{\text{Observed Accuracy-Expected Accuracy}}{1 - \text{Expected Accuracy}} \tag{22}$$

### 4.2. Baseline State-of-the-Art CNN Image Classification Architecture

Many of the baseline CNN models we selected for comparison were carefully selected for various reasons, including being winning entries to image analysis and classification competitions and are state of the art and/or performed well on similar problems. They include:

DenseNet201 [58]: This is a 201-layer CNN in which each layer is connected to every other layer in a feedforward manner to eliminate the vanishing gradient problem, enhance feature propagation, promote reuse of features, and drastically reduce the number of parameters. DenseNet is based on the idea that convolutional networks can be more accurate and efficient to train if they have shorter connections between the layers near the input and the layers near the output. We selected DenseNet 201 because it was utilized in prior work [59] as a feature extractor for deep hybrid architectures for binary classification of breast cancer images.

SqueezNet [60]: This is a lightweight CNN that employs various design strategies that reduce the number of parameters, particularly with use of fire modules which "squeeze" parameters using $1 \times 1$ convolutions for the network to carry fewer parameters. The problem of storage efficiency and speed of models for prediction was solved using a technique known as model compression, which it accomplished by: (i) compressing the perspective of model weight values, and (ii) compressing the perspective of network architecture. SqueezeNet was selected because it was previously utilized for deep feature extraction and classification of breast ultrasound images [61].

ShuffleNet [62]: This is a convolutional neural network specifically designed for mobile devices with low processing power. The architecture uses two new operations, pointwise group convolution and channel shuffle, to reduce computation costs while preserving accuracy. ShuffleNet was selected as a baseline because it was utilized for breast cancer classification in prior work [63].

### 4.3. Experiments

In this section, experiments to rigorously evaluate our proposed BoDMCF approach using the BreakHis dataset of histopathological breast cancer images [27] that is summarized in Table 2 are described. The classification task was performed by fine-tuning (transfer learning) the CNN models that were previously pre-trained on the ImageNet dataset and on the BreakHis dataset. Various hyperparameters shown in Table 6 were determined using grid search and specified, followed by pre-processing, training, and validation of histopathological images. Test images were then provided as inputs to the trained models. The fine-tuned, pre-trained CNN models were used to extract features at four resolutions, which were pooled to form the BoDMCF that was then classified using SVM. The classifier performance was evaluated with ten-fold cross-validation with a cross-validation error of 0.0462.

**Table 6.** Optimal hyperparameters used for pre-trained models.

| Hyperparameter | Value |
| --- | --- |
| Train–Test ratio | 70:30 |
| optimization algorithm | stochastic gradient descent |
| activation function | ReLu |
| Mini Batch Size | 20 |
| Max Epochs | 30 |
| Initial Learn Rate | 0.00125 |
| Learn-Rate Drop Factor | 0.1 |
| Learn-Rate Drop Period | 20 |

Experiment: train-test curves: Figure 10 shows sample train–test curves we generated during training of the EfficientNetb0 model demonstrating model convergence after about 200 epochs.

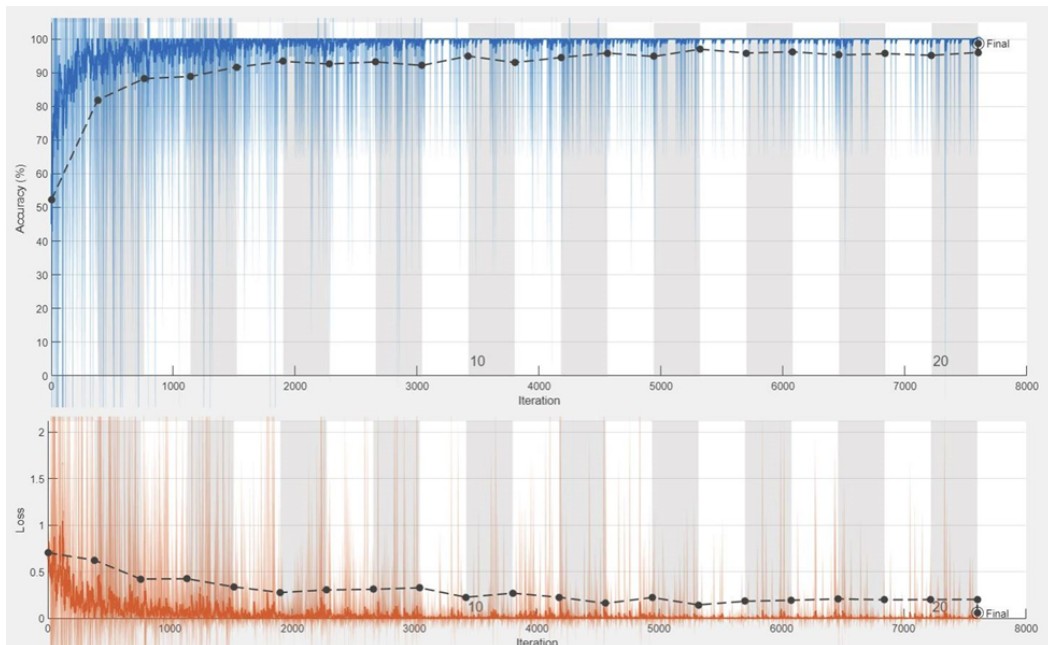

**Figure 10.** Efficientnetbo train–test performance. The small difference between the training and test loss curves demonstrate that there is no overfitting.

Experiment: binary classification (benign vs. malignant) of individual magnifications ($40\times$, $100\times$, $200\times$, and $400\times$) of histopathological breast cancer images using baseline models with model parameters (weights) determined by pre-training on ImageNet weights: The goal of this experiment was to establish baseline performance of individual state-of-the-art CNN image classification models (ResNet18, InceptionV3, InceptionResnetV2, DenseNet201, ResNet50, EfficientNetB0, SqueezeNet, and ShuffleNet) using weights determined via pre-training on ImageNet (no fine-tuning on the BreakHis dataset). Classification was at individual image resolution with no pooling of features to create the BoDMCF. Our goal was to eventually demonstrate that pooling multiple resolutions of features to create our BoDMCF approach outperforms these powerful baselines that perform classification on single image resolutions. The results of this experiment are shown in Table 7. Except for the precision metric (ResNet18 on $200\times$ magified image hasthe highest precision), SqueezeNet performed best on all other metrics (accuracy, F1 score, recall, AUC, Kappa, and MCC). These results suggest that visual attributes that most clearly distinguish malignant tumors from benign ones are most observable at $100\times$ magnification and that the SqueezeNet neural networks model outperforms all other baseline models when model weights learned from ImageNet during pre-training (no fine-tuning on the BreakHis dataset) are utilized.

**Table 7.** Binary (benign vs. malignant) classifier performance: comprehensive table of metrics for classifying various magnification levels of histopathological images using baseline CNN models with pre-trained weight from ImageNet (no fine-tuning of weights on the BreakHis dataset).

| Model | Accuracy | Precision | F1-Score | Recall | AUC | Kappa | MCC |
|---|---|---|---|---|---|---|---|
| 40× Renent18 | 0.9064 | 0.8047 | 0.8607 | 0.9251 | 0.9115 | 0.8667 | 0.7950 |
| 40× InceptionResnetV2 | 0.7843 | 0.6021 | 0.7261 | 0.9144 | 0.8197 | 0.6924 | 0.5937 |
| 40× InceptionV3 | 0.8679 | 0.8418 | 0.7710 | 0.7112 | 0.8252 | 0.8247 | 0.6839 |
| 40× Densenet201 | 0.8963 | 0.9433 | 0.8110 | 0.7112 | 0.8459 | 0.8629 | 0.7555 |
| 40× Resnet50 | 0.9381 | 0.9261 | 0.8981 | 0.8717 | 0.9200 | 0.9135 | 0.8545 |
| 40× Efficientnetbo | 0.8311 | 0.7389 | 0.7248 | 0.7112 | 0.7984 | 0.7749 | 0.6033 |
| 40× Squeezenet | 0.9281 | 0.8429 | 0.8917 | 0.9465 | 0.9331 | 0.8970 | 0.8413 |
| 40× Shufflenet | 0.8428 | 0.8252 | 0.7152 | 0.6310 | 0.7851 | 0.7972 | 0.6197 |
| 100× Renent18 | 0.9327 | 0.8995 | 0.8901 | 0.8808 | 0.9184 | 0.9060 | 0.8417 |
| 100× InceptionResnetV2 | 0.8173 | 0.7582 | 0.6705 | 0.6010 | 0.7576 | 0.7669 | 0.5535 |
| 100× InceptionV3 | 0.9022 | 0.9342 | 0.8232 | 0.7358 | 0.8563 | 0.8701 | 0.7673 |
| 100× Densenet201 | 0.9183 | 0.9329 | 0.8571 | 0.7927 | 0.8836 | 0.8892 | 0.8056 |
| 100× Resnet50 | 0.9119 | 0.8670 | 0.8556 | 0.8446 | 0.8933 | 0.8783 | 0.7924 |
| 100× Efficientnetbo | 0.8526 | 0.8301 | 0.7341 | 0.6580 | 0.7989 | 0.8087 | 0.6422 |
| 100× Squeezenet | **0.9455** | 0.8630 | **0.9175** | **0.9793** | **0.9548** | **0.9216** | **0.8809** |
| 100× Shufflenet | 0.8606 | 0.7454 | 0.7873 | 0.8342 | 0.8533 | 0.8075 | 0.6865 |
| 200× Renent18 | 0.8791 | **0.9831** | 0.7607 | 0.6203 | 0.8078 | 0.8460 | 0.7178 |
| 200× InceptionResnetV2 | 0.8692 | 0.9030 | 0.7539 | 0.6471 | 0.8079 | 0.8313 | 0.6853 |
| 200× Inceptionv3 | 0.8957 | 0.9079 | 0.8142 | 0.7380 | 0.8522 | 0.8611 | 0.7504 |
| 200× Densenet201 | 0.8891 | 0.9000 | 0.8012 | 0.7219 | 0.8430 | 0.8531 | 0.7340 |
| 200× Resnet50 | 0.8642 | 0.7561 | 0.7908 | 0.8289 | 0.8545 | 0.8128 | 0.6922 |
| 200× Efficientnetbo | 0.8245 | 0.7396 | 0.7022 | 0.6684 | 0.7815 | 0.7704 | 0.5798 |
| 200× Squeezenet | 0.9205 | 0.9017 | 0.8667 | 0.8342 | 0.8967 | 0.8907 | 0.8114 |
| 200× Shufflenet | 0.8526 | 0.8451 | 0.7295 | 0.6417 | 0.7945 | 0.8099 | 0.6421 |
| 400× Renent18 | 0.8681 | 0.9127 | 0.7616 | 0.6534 | 0.8118 | 0.8273 | 0.6918 |
| 400× InceptionResnetV2 | 0.8168 | 0.7262 | 0.7093 | 0.6932 | 0.7844 | 0.7545 | 0.5760 |
| 400× InceptionV3 | 0.7253 | 0.5422 | 0.6901 | 0.9489 | 0.7839 | 0.5988 | 0.5351 |
| 400× Densenet201 | 0.8626 | 0.8686 | 0.7604 | 0.6761 | 0.8137 | 0.8183 | 0.6765 |
| 400× Resnet50 | 0.8846 | 0.9185 | 0.7974 | 0.7045 | 0.8374 | 0.8460 | 0.7311 |
| 400× Efficientnetbo | 0.8278 | 0.6971 | 0.7552 | 0.8239 | 0.8268 | 0.7585 | 0.6290 |
| 400× Squeezenet | 0.9231 | 0.9524 | 0.7947 | 0.6818 | 0.8328 | 0.8499 | 0.7383 |
| 400× Shufflenet | 0.8736 | 0.9350 | 0.7692 | 0.6534 | 0.8159 | 0.8346 | 0.7068 |

Experiment: binary classification (benign vs. malignant) using features extracted from individual magnifications (40×, 100×, 200×, and 400×) of histopathological breast cancer images by baseline CNN models fine-tuned on the BreakHis dataset, which are then classified using SVM: The goal of this experiment was to demonstrate the power of pooling multiple resolutions of deep CNN features. Specifically, we benchmarked the performance of deep features extracted at individual magnifications using state-of-the-art fine-tuned CNN image classification models (ResNet18, InceptionV3, InceptionResnetV2, DenseNet201, ResNet50, EfficientNetB0, SqueezeNet, and ShuffleNet) without pooling multiple magnifications into a single BoDMCF representation as we proposed. The results of this experiment are shown in Table 8. Except for the precision metric (ResNet50 on the 40× magnified image has the highest precision), DenseNet201 performed best on all other metrics (accuracy, F1 score, recall, AUC, Kappa, and MCC). These results suggest that when utilized as feature extractors, visual attributes that most clearly distinguish malignant tumors from benign ones are most observable at 40× magnification and that using the DenseNet201 with fine-tuning on the BreakHis dataset outperforms all other baselines as a feature extractor.

**Table 8.** Binary (benign vs. malignant) classifier performance: comprehensive table of metrics for deep features extracted from different individual magnifications of histopathological images using fine-tuned CNN models that are classified using SVM (no pooling of features to create a bag (BoDMCF).

| Model | Accuracy | Precision | F1-Score | Recall | AUC | Kappa | MCC |
|---|---|---|---|---|---|---|---|
| 40× Resnet18 | 0.9064 | 0.8619 | 0.8478 | 0.8342 | 0.8867 | 0.8706 | 0.7804 |
| 40× InceptionResnetV2 | 0.9214 | 0.8723 | 0.8746 | 0.8770 | 0.9093 | 0.8899 | 0.8174 |
| 40× InceptionV3 | 0.9348 | 0.9353 | 0.8908 | 0.8503 | 0.9118 | 0.9095 | 0.8464 |
| 40× Densenet201 | **0.9615** | 0.9409 | **0.9383** | **0.9358** | **0.9545** | **0.9451** | **0.9104** |
| 40× Resnet50 | 0.9548 | **0.9546** | 0.9257 | 0.8984 | 0.9395 | 0.9364 | 0.8941 |
| 40× Efficientnetbo | 0.9448 | 0.9326 | 0.9096 | 0.8877 | 0.9293 | 0.9225 | 0.8705 |
| 40× Squeezenet | 0.8395 | 0.7301 | 0.7496 | 0.7701 | 0.8205 | 0.7817 | 0.6324 |
| 40× Shufflenet | 0.8746 | 0.8146 | 0.7945 | 0.7754 | 0.8476 | 0.8297 | 0.7048 |
| 100× Renent18 | 0.9199 | 0.8743 | 0.8698 | 0.8653 | 0.9048 | 0.8887 | 0.8119 |
| 100× InceptionResnetV2 | 0.9006 | 0.8466 | 0.8377 | 0.8290 | 0.8809 | 0.8634 | 0.7662 |
| 100× InceptionV3 | 0.9022 | 0.8743 | 0.8338 | 0.7969 | 0.8729 | 0.8671 | 0.7663 |
| 100× Densenet201 | 0.9151 | 0.8763 | 0.8602 | 0.8446 | 0.8956 | 0.8828 | 0.7995 |
| 100× Resnet50 | 0.9391 | 0.8894 | 0.9031 | 0.9171 | 0.9330 | 0.9140 | 0.8589 |
| 100× Efficientnetb0 | 0.9295 | 0.8821 | 0.8866 | 0.8912 | 0.9189 | 0.9012 | 0.8355 |
| 100× Squeezenet | 0.8542 | 0.7713 | 0.7612 | 0.7513 | 0.8258 | 0.8041 | 0.6563 |
| 100× Shufflenet | 0.9022 | 0.8511 | 0.8399 | 0.8290 | 0.8820 | 0.8657 | 0.7697 |
| 200× Renent18 | 0.8940 | 0.8090 | 0.8342 | 0.8610 | 0.8849 | 0.8527 | 0.7572 |
| 200× InceptionResnetV2 | 0.9421 | 0.9000 | 0.9072 | 0.9144 | 0.9344 | 0.9182 | 0.8651 |
| 200× InceptionV3 | 0.8990 | 0.8663 | 0.8301 | 0.7968 | 0.8708 | 0.8627 | 0.7598 |
| 200× Densenet201 | 0.9400 | 0.8989 | 0.8767 | 0.8556 | 0.9062 | 0.8968 | 0.8239 |
| 200× Resnet50 | 0.9338 | 0.9016 | 0.8919 | 0.8824 | 0.9196 | 0.9075 | 0.8443 |
| 200× Efficientnetbo | 0.9437 | 0.9048 | 0.9096 | 0.9144 | 0.9356 | 0.9206 | 0.8687 |
| 200× Squeezenet | 0.9056 | 0.8385 | 0.8496 | 0.8610 | 0.8933 | 0.8689 | 0.7810 |
| 200× Shufflenet | 0.9040 | 0.8817 | 0.8371 | 0.7968 | 0.8744 | 0.8695 | 0.7712 |
| 400× Renent18 | 0.8516 | 0.7811 | 0.7652 | 0.7500 | 0.8250 | 0.7977 | 0.6571 |
| 400× InceptionResnetV2 | 0.8663 | 0.8160 | 0.7847 | 0.7557 | 0.8373 | 0.8176 | 0.6890 |
| 400× InceptionV3 | 0.9011 | 0.8466 | 0.8466 | 0.8466 | 0.8868 | 0.8609 | 0.7736 |
| 400× Densenet201 | 0.8900 | 0.8650 | 0.8319 | 0.8011 | 0.8708 | 0.8555 | 0.7575 |
| 400× Resnet50 | 0.9176 | 0.8659 | 0.8732 | 0.8807 | 0.9079 | 0.8828 | 0.8123 |
| 400× Efficientnetbo | 0.9176 | 0.8503 | 0.9034 | 0.8760 | 0.9139 | 0.8819 | 0.8152 |
| 400× Squeezenet | 0.8626 | 0.7919 | 0.7851 | 0.7784 | 0.8406 | 0.8109 | 0.6842 |
| 400× Shufflenet | 0.8828 | 0.8111 | 0.8202 | 0.8295 | 0.8688 | 0.8359 | 0.7334 |

Experiment: binary classification (benign vs. malignant) of a combined pool of all four magnifications (40×, 100×, 200×, and 400×) of histopathological breast cancer images using baseline models with model parameters (weights) determined by pre-training on ImageNet: The main difference with our proposed approach is that, while all four magnifications were pooled in this experiment, only a single CNN pre-trained model (one of ResNet18, InceptionV3, InceptionResnetV2, DenseNet201, ResNet50, EfficientNetB0, SqueezeNet. and ShuffleNet) was used for classifying the pool of images at a time. In contrast, our proposed approach extracts features using an ensemble of three CNN models (ResNet18, InceptionV3, and ResNetInceptionV2). The results of this experiment are shown in Table 9. Except for the precision metric (DenseNet201 has the highest precision), SqueezeNet performed best on all other metrics (accuracy, F1 score, re- call, AUC, Kappa, and MCC). These results suggest the SqueezeNet neural networks architecture outperforms all other baselines on a multi-resolution bag of features when model weights learned from ImageNet during pre-training are utilized.

**Table 9.** Binary (benign vs. malignant) classifier performance: comprehensive table of metrics for classifying a combined pool of all histopathological image magnifications ($40\times$, $100\times$, $200\times$, and $400\times$) extracted using a single state-of-the-art baseline CNN model.

| Model | Accuracy | Precision | F1-Score | Recall | AUC | Kappa | MCC |
|-------|----------|-----------|----------|--------|-----|-------|-----|
| Renent18 | 0.8896 | 0.9382 | 0.7975 | 0.6935 | 0.8363 | 0.8548 | 0.7396 |
| InceptionResnetV2 | 0.8609 | 0.8405 | 0.7559 | 0.6868 | 0.8136 | 0.8169 | 0.6666 |
| InceptionV3 | 0.8845 | 0.9211 | 0.7896 | 0.6909 | 0.8319 | 0.8482 | 0.7262 |
| Densenet201 | 0.9166 | **0.9535** | 0.8529 | 0.7715 | 0.8772 | 0.8872 | 0.8043 |
| Resnet50 | 0.9048 | 0.8960 | 0.8383 | 0.7876 | 0.8729 | 0.8704 | 0.7745 |
| Efficientnetb0 | 0.8281 | 0.8590 | 0.6634 | 0.5403 | 0.7499 | 0.7857 | 0.5827 |
| Squeezenet | **0.9460** | 0.9195 | **0.9201** | **0.9207** | **0.9419** | **0.9287** | **0.8835** |
| Shufflenet | 0.8643 | 0.8879 | 0.7500 | 0.6492 | 0.8059 | 0.8241 | 0.6752 |

Experiment: binary (benign vs. malignant) classifier performance: comprehensive table of metrics for features extracted from a pooled combination of all four magnifications ($40\times$, $100\times$, $200\times$, and $400\times$) of histopathological images using baseline CNN models that are classified using SVM The main difference with our proposed approach is that, while all four magnifications were pooled in this experiment, features were extracted using only a single CNN pre-trained model (one of ResNet18, InceptionV3, InceptionResnetV2, DenseNet201, ResNet50, EfficientNetB0, SqueezeNet, and ShuffleNet) at a time. In contrast, our proposed approach extracts features using an ensemble of three CNN models (ResNet18, InceptionV3, and ResNetInceptionV2). The results of this experiment are shown in Table 10. Except for the precision metric (DenseNet50 has the highest precision), EfficientNetB0 performed best on all other metrics (accuracy, F1 score, recall, AUC, Kappa, and MCC). These results suggest the EfficientNetB0 neural networks architecture outperforms all other baselines as a deep feature extractor from a pool of multiple magnifications of histopathological images.

**Table 10.** Binary (benign vs. malignant) classifier performance: comprehensive table of metrics for features extracted from a pooled combination of all four magnifications ($40\times$, $100\times$, $200\times$, and $400\times$) of histopathological images using baseline CNN models that are classified using SVM.

| Model | Accuracy | Sensitivity | AUC | Fscore | TPR | FPR | MCC | Kappa | Prec | Spec |
|-------|----------|-------------|-----|--------|-----|-----|-----|-------|------|------|
| Densenet201 | 0.9815 | 0.9677 | 0.9777 | 0.9704 | 0.9677 | 0.0123 | 0.9569 | 0.8458 | 0.9730 | 0.9877 |
| Resnet50 | **0.9899** | **0.9892** | **0.9897** | **0.9840** | **0.9892** | **0.0098** | **0.9766** | 0.8410 | **0.9787** | **0.9902** |
| Efficientnetb0 | 0.9836 | 0.9718 | 0.9804 | 0.9737 | 0.9718 | 0.0110 | 0.9618 | 0.8447 | 0.9757 | 0.9890 |
| InceptionResnetV2 | 0.9823 | 0.9866 | 0.9835 | 0.9722 | 0.9866 | 0.0196 | 0.9594 | 0.8439 | 0.9582 | 0.9804 |
| InceptionV3 | 0.9836 | 0.9798 | 0.9826 | 0.9739 | 0.9798 | 0.0147 | 0.9620 | 0.8440 | 0.9681 | 0.9853 |
| Renent18 | 0.9777 | 0.9798 | 0.9783 | 0.9649 | 0.9798 | 0.0233 | 0.9488 | 0.8461 | 0.9505 | 0.9767 |
| Shufflenet | 0.9794 | 0.9691 | 0.9766 | 0.9671 | 0.9691 | 0.0160 | 0.9521 | 0.8464 | 0.9652 | 0.9840 |
| Squeezenet | 0.9659 | 0.9664 | 0.9660 | 0.9467 | 0.9664 | 0.0344 | 0.9220 | **0.8512** | 0.9277 | 0.9656 |

Results of Our BoDMCF approach: Our approach has two key distinctions with the baseline approaches presented thus far. First, we extract features from all four magnifications of histopathological images, which are then pooled into a BoDMCF. Secondly, we use multiple (three) state-of-the-art CNN models (ResNet-50, InceptionV3, and Efficientnet-b0) as feature extractors. The results of our approach, which are shown in Table 10 for individual networks and Table 11, demonstrate that our approach outperforms the baseline approaches. Figure 11 shows samples of test images with their predicted labels from our

proposed method. Finally, to demonstrate that the difference in performance between our BoDMCF approach and other ensemble baselines is was statistically significant, we performed the Nemenyi post hoc test [64]. At a confidence level a = 0.05, the critical distance (CD) is 1.2536.

**Table 11.** Results of our proposed approach with features extracted from all four histopathological image magnifications ($40\times$, $100\times$, 200, $\times$ and $400\times$) by three state-of-the-art CNN models (ResNet-50-, InceptionV3, and EfficientNet-b0). The effects of the number of features used on model performance are also shown. The 2- and 3-model combinations were based on best performing single model performance in Table 9. Accuracy achieved by prior breast cancer binary classification work are also shown in the bottom Table 12.

| F. Extractors | Size | Bytes | Acc. | Sens. | Spec. | AUC | F1-Sc. | TPR | FPR | MCC | Kappa | Prec. |
|---|---|---|---|---|---|---|---|---|---|---|---|---|
| Efficientnetb0 $ Resnet50 | Train_features 5536 × 3328 Test_features 2373 × 3328 | Train_features 73,695,232 Test_features 31,589,376 | 0.9966 | 0.9987 | 0.9662 | 0.9972 | 0.9946 | 0.9987 | 0.0043 | 0.9922 | 0.8378 | 0.9907 |
| Efficientnetb0 $ Inception-V3 | Train_features 5536 × 2816 Test_features 2373 × 2816 | Train_features 62,357,504 Test_features 26,729,472 | 0.9962 | 0.9933 | 0.9699 | 0.9352 | 0.9939 | 0.9933 | 0.0025 | 0.9912 | **0.9352** | 0.9946 |
| Resnet50 $ Inception-V3 | Train_features 5536 × 3584 Test_features 2373 × 3584 | Train_features 79,364,096 Test_features 34,019,328 | 0.9979 | 0.9960 | 0.9705 | 0.9974 | 0.9966 | 0.9960 | 0.0012 | 0.9951 | 0.8375 | 0.9973 |
| BoDMCF | Train_features 5536 × 4864 Test_features 2373 × 4864 | Train_features 107,708,416 Test_features 46,169,088 | **0.9992** | **0.9987** | **0.9797** | **0.9990** | **0.9987** | **0.9987** | **0.0006** | **0.9980** | 0.8368 | **0.9987** |

**Table 12.** Accuracy achieved by prior breast cancer binary classification work are also shown in the bottom table.

| Authors | Models | Accuracy |
|---|---|---|
| M. Amrane [14] | Naive Bayes (NB) k-nearest neighbor (KNN) | 97.51% for KNN and 96.19% for NB |
| S. H. Kassani, M. J. Wesolowski, and K. A. Schneider [29] | VGG19, MobileNet, and DenseNet | 98.13% |
| F. A. Spanhol, L. S. Oliveira, C. Petitjean, and L. Heutte [31] | Ensemble models | 85.6% |
| Kowal et al. [32] | Deep learning model | 92.4% |
| A. Al Nahid, M. A. Mehrabi, and Y. Kong [36] | CNN, LSTM, K-means clustering, Mean-Shift clustering and SVM | 96.0% |
| **Our Proposed Approach** | **BoDMCF + SVM** | **99.92%** |

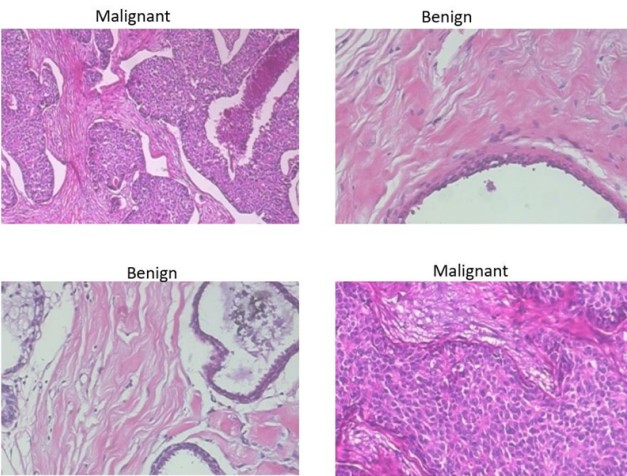

**Figure 11.** Four sample test images with their predicted labels from our proposed algorithm.

Experiment: ROC Curves The receiver operating characteristic (ROC) curve, shown in Figure 12 for our approach, is a graphical plot that shows the diagnostic ability of a binary classifier as its discrimination threshold is varied. In simple terms, the ROC curve plots our approaches FPR vs. its TPR. The ROC curve is almost a perfect right angle at the top left corner, demonstrating that our proposed approach achieves excellent FPR and TPR.

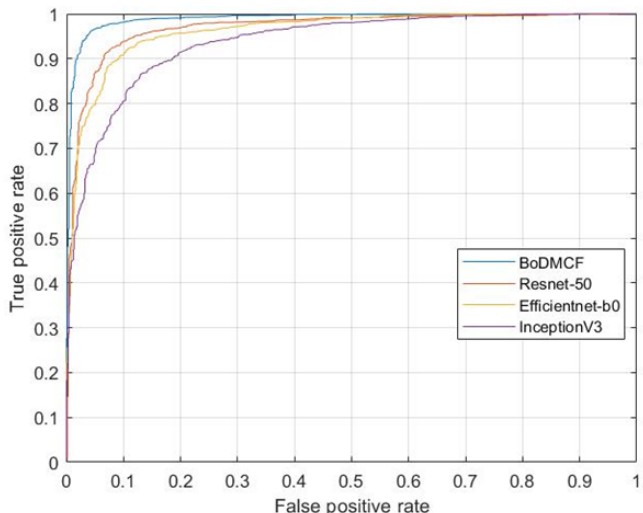

**Figure 12.** ROC Curves for our Approach.

Experiment: Confusion Matrix: To evaluate which classes were confounded by other classes, we analyzed the confusion matrix. The confusion matrix of the top performing technique is presented in Figure 13. The columns correspond to the targeted class, and the rows correspond to the output class (anticipated class). The diagonal cells match with observations that are rightly classified. The off-diagonal cells refer to incorrect classifications. The percentage of the overall number of observations and the number of observations in every cell is also presented. The column on the extreme right displays the proportions of incorrect (red color) and correct (green color) classifications that were predicted. These metrics are referred to as the false discovery rate and the positive predictive value. While the lowest row indicates the percentages of incorrect and correct classifications, and these metrics are referred to as false negative rate (FNR) and true positive rate (TPR), the cell in the bottom-most right shows the general precision. A column-normalized column summary displays the percentages of incorrectly and correctly classified observations for every predicted class. A row-standardized row summary exhibits the percentages of incorrectly

and correctly classified observations for every true class. In the confusion matrix, most of the results fall on the leading diagonal with very few off the diagonal, which demonstrates that the proposed approach did not confuse the benign and malignant classes.

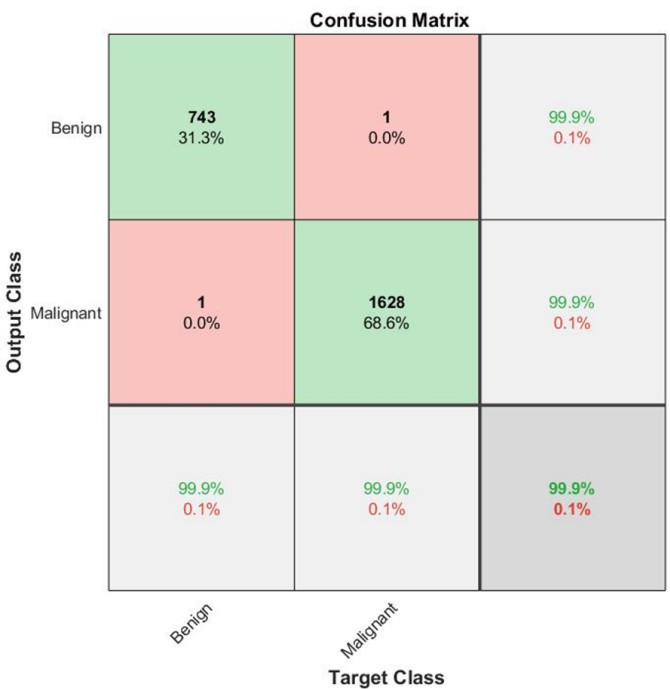

**Figure 13.** Confusion matrix displaying the performance of the proposed BoDMCF approach.

Experiment: classifying the BoDMCF representation using different machine learning classifiers: The goal of this experiment was to compare the performance of the support vector machines (SVM) with other traditional machine learning (ML) classifiers for the task of classifying the BoDMCF representation into target labels of "Benign" and "Malignant". Results in Table 13 show that SVM outperformed all other ML classifiers for this binary classification task. This is likely because SVM is well-known to perform well on binary classification tasks.

**Table 13.** Results of classifying the BoDMCF with various machine learning (ML) classifiers.

| ML Classifier | Acc. | Sens. | Spec. | AUC | F1 | TPR | FPR | MCC | Kappa | Prec. |
|---|---|---|---|---|---|---|---|---|---|---|
| Binary Decision Classifier | 0.9660 | 0.9449 | 0.9761 | 0.9605 | 0.9462 | 0.9449 | 0.0239 | 0.9216 | 0.8529 | 0.9474 |
| Linear Discriminant Analysis (LDA) | 0.9900 | 0.9892 | 0.9902 | 0.9897 | 0.9840 | 0.9892 | 0.0098 | 0.9766 | 0.8410 | 0.9787 |
| Generalized Additive Model | 0.9660 | 0.9489 | 0.9742 | 0.9616 | 0.9464 | 0.9489 | 0.0258 | 0.9218 | 0.8526 | 0.9439 |
| Gradient Boosted Machines (GBM) | 0.9870 | 0.9812 | 0.9890 | 0.9851 | 0.9786 | 0.9812 | 0.0110 | 0.9687 | 0.8429 | 0.9759 |
| K-Nearest Neighbor (KNN) | 0.9800 | 0.9960 | 0.9724 | 0.9842 | 0.9686 | 0.9960 | 0.0276 | 0.9545 | 0.8440 | 0.9427 |
| Naive Bayes (NB) | 0.9730 | 0.9919 | 0.9638 | 0.9779 | 0.9578 | 0.9919 | 0.0362 | 0.9388 | 0.8468 | 0.9260 |
| Support Vector Machines (SVM) | **0.9992** | **0.9987** | **0.9797** | **0.9990** | **0.9987** | **0.9987** | **0.0006** | **0.9980** | **0.8368** | **0.9987** |

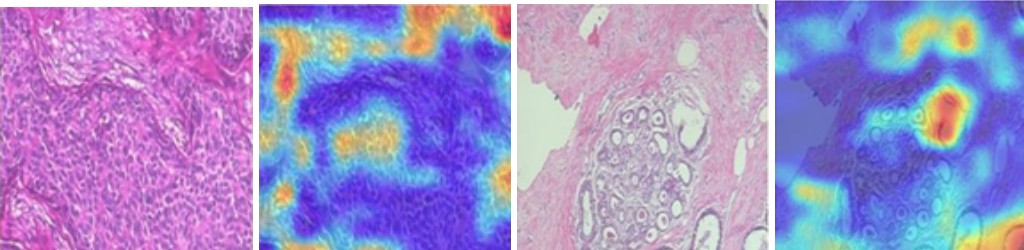

**Figure 14.** Sample heatmaps of regions of interest generated by grad-cam (left pair = malignant original image and grad-cam heatmap, right pair = benign original image and grad-cam heatmap.

Experiment: CNN model interpretability using grad-cam [65] The goal of this experiment was to ensure that the breast cancer classification model focused on the appropriate regions of the image when analyzing the image. Grad-cam computes the gradient of the ranking score in relation to the CNN characteristics map, highlighting the specific ROIs based on the greatest gradient score. Grad-cam computes the gradients with respect to feature maps of a convolutional layer, which are then global-average-pooled to obtain the importance weights $\alpha_k^c$; $\alpha_k^c$ represents a partial linearization of the deep network downstream from $A$, capturing the importance of feature map $k$ for a target class $c$

$$\alpha_k^c = \overbrace{\frac{1}{Z}\sum_i\sum_j}^{\text{global average pooling}} \underbrace{\frac{\partial y^c}{\partial A_{ij}^k}}_{\text{gradients via backprop}} \tag{23}$$

$\frac{\partial y^c}{\partial A_{ij}^k}$ is the gradient of the score for class $c$, $y^c$, with respect to feature maps $A^k$ of a convolutional layer. A grad-cam heatmap is then generated as a weighted combination of forward activation feature maps, but followed by a ReLU activation function

$$L_{\text{Grad-CAM}}^c = \text{Re}\,LU\,\underbrace{\left(\sum_k \alpha_k^c A^k\right)}_{\text{linear combination}} \tag{24}$$

where $L_{\text{Grad-CAM}}^c$ is the class-discriminative localization map. Grad-cam was applied to produce a coarse localized map highlighting the most important ROIs in the histopathological images to classify the images as benign or malignant. Sample grad-cam results are shown in Figure 14.

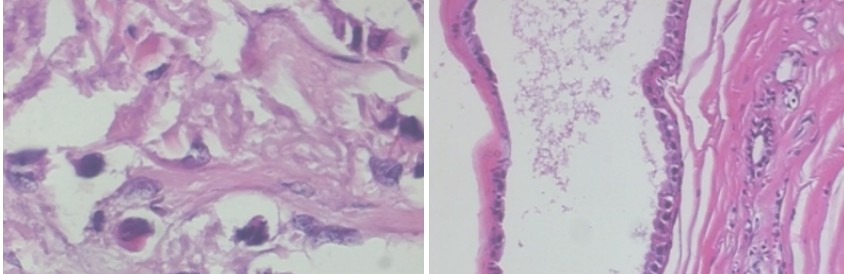

**Figure 15.** Sample misclassified images. **Left** is a misclassified malignant image, and **right** is a misclassified benign image.

Experiment: analysis of misclassified images: The objective of this experiment was to discover reasons behind model misclassifications, which could be addressed either to improve this manuscript or in future work. Misclassifications resulted for benign images that looked similar to malignant images or vice versa. One example each of misclassified

malignant and benign histopathological images, respectively, from the Breakhis dataset is shown in Figure 15. The outline and uniformity of the texture differences in the benign image are comparable to those in a malignant image. There is less dispersion of cells in misclassified malignant images than in ordinary malignant images. Consequently, the cells appear benign, resulting in misclassification. Benign histopathology images usually have fewer dispersed cells and only a few spreads elsewhere.

## 5. Discussion

Through rigorous experimentation, as shown in Table 11, we demonstrated that the proposed BoDMCF approach outperforms a comprehensive set of baselines as well as the prior state-of-the-art methods (Table 12) for binary classification of histopathological images. Our results also demonstrate that all key components of our approach contribute non-trivially to its superior results, including:

Transfer learning by pre-training on a large image repository (ImageNet) with fine-tuning on the BreakHis breast cancer image dataset that enables the CNN feature extractors models to learn a robust image representation from the large image repository. Fine-tuning on the BreakHis breast cancer dataset transfers the learned intelligence to the task of analyzing and classifying breast cancer. This conclusion is evident by comparing results in Tables 7 (pre-training with no fine-tuning) and 8 (pre-training with fine-tuning).

Using an ensemble of CNNs for deep feature extractors achieves superior performance to using any single pre-trained CNN for feature extraction, which is evident by comparing results in Tables 10 and 11. In fact, as shown by the results in Table 11, we also show that the three specific state-of-the-art CNN models (ResNet-50, InceptionV3, and Efficientnet-b0) discovered through extensive experimentation and utilized for feature extraction, outperform other CNN combinations and ensembles. Intuitively, each CNN extracts slightly different image features. Feature extraction using multiple CNNs combines these different features into a superset of features that outperforms features extracted from any single CNN.

Extracting deep features for four magnifications ($40\times$, $100\times$, $200\times$, and $400\times$) of histopathological images that are then pooled into a BoDMCF, is important as the visual attributes that distinguish malignant from benign tumors may be most discernable at different resolutions. This conclusion is evident because the results of the pooled, multiresolution BoDMCF features (Table 13) outperform results of classifying deep features extracted from any individual single resolution as shown in Table 8.

Global pooling of multiresolution features to create a bag (BoDMCF0) is an essential step that also enables downstream classification using traditional machine learning algorithms such as SVM. Deep BoDMCFs are a powerful representation, which had the best performance for all combinations of CNN models explored in this study as shown in Table 13. The proposed technique of using BoDMCF features, pooled and classified using SVM, outperformed single CNN model approaches in Table 10.

SVM outperformed all other traditional ML classification algorithms for classifying the BoDMCF into malignant and benign target classes as shown in Table 13. We believe that this is because SVM's maximal margin hyperplane determination approach performs well on binary classification.

Limitations of this work and potential future work: The results acquired show that very significant classification performance can be achieved. While our proposed approach is shown to perform well on the BreakHis dataset, one of the most widely distributed histopathological images hosted on the public domain, some limitations can be addressed in future work. Firstly, extending the dataset to include more images from more magnifications could yield more robust classifiers before deployment for use in hospitals. Secondly, we used three existing deep models. In future, fusing deeper models could yield better performance. Third, we would like to validate our results on other histopathological breast cancer datasets. Finally, implementing our methods on mobile devices can be a promis-

ing direction that facilitates deployment in under-resourced environments such as third world countries.

## 6. Conclusions

We have proposed an automatic classification method for breast cancer histopathological images into malignant vs. benign categories. Particularly, we have shown that a deep BoDMCF feature extraction from multiple magnifications ($40\times$, $100\times$, $200\times$, and $400\times$) of histopathological images using three state-of-the-art pre-trained CNN models (ResNet-50, Inception-v3, and EfficientNet-b0) with pooling and classification using SVM, can also be leveraged for binary (malignant vs. benign) breast cancer classification. Moreover, combining deep rich features from various global average pooling layers of various pre-trained convolutional deep models was shown to yield improved classification performance. In rigorous evaluation experiments, our deep BoDMCF feature approach with global pooling achieved an average accuracy of 99.92% for the classification task, sensitivity of 0.9987, specificity (or recall) of 0.9797, positive prediction value (PPV) or precision of 0.99870, F1-Score of 0.9987, MCC of 0.9980, Kappa of 0.8368, and AUC of 0.9990 on the BreaKHis dataset [27]. Our deep BoMCF approach outperforms state-of-the-art CNN baselines including ResNet18, InceptionV3, DenseNet201, EfficientNetb0, SqueezeNet, and ShuffleNet when classifying any of the individual resolutions ($40\times$, $100\times$, $200\times$ or $400\times$) or when SVM is used to classify a BoMCF extracted using any single pre-trained CNN model. The high accuracy, sensitivity, PPV, and F1 score achieved by our approach is extremely encouraging and could be useful in supporting the work of health practitioners in low-resource settings with few experts. However, before deployment, a careful validation study and comparison of our model's performance to human experts needs to be conducted. In future work, combining several other image magnifications using emerging CNN models could yield even better breast cancer classification models.

**Author Contributions:** D.C., E.A. and W.S. conceived the presented idea; J.O. and S.A. were involved in the conceptualization and supervision of the research; D.C. and E.A. developed the theory and performed the computations; E.A. and W.S. verified the analytical methods; E.A. and W.S. encouraged D.C. to investigate the algorithm used for this work and supervised the findings of this work. All authors discussed the results and contributed to the final manuscript. All authors have read and agreed to the published version of the manuscript.

**Funding:** This research received no external funding.

**Institutional Review Board Statement:** The study utilized a publicly available, de-identified dataset. Hence, IRB approval was not required.

**Informed Consent Statement:** The study utilized a publicly available, de-identified dataset. Hence, IRB approval and an informed consent was not required.

**Data Availability Statement:** The study utilized BreakHis, a publicly available, de-identified breast cancer dataset that is available at https:\\web.inf.ufpr.br\vri\databases\breast-cancer-histopathological-database-breakhis (accessed on 1 October 2022).

**Conflicts of Interest:** The authors declare no conflict of interest.

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
