# Peer review of "Breast Cancer Tumor Classification Using a Bag of Deep Multi-Resolution Convolutional Features"

_informatics, doi:10.3390/informatics9040091_

Round 1

Reviewer 1 Report

In this manuscript, the author proposed a bag of deep multi-resolution convolutional features, called BoDMCF, as a feature extraction technique. The features were then sent to classify as a breast cancer tumor or not (benign and malignant) using various machine learning, such as binary decision classifier, LDA, generalized additive model, GBM, KNN, NB, and SVM. The results showed that the SVM technique achieved 99.92% accuracy and outperformed other machine learning methods. In comparison, the linear discriminant analysis (LDA) achieved 99.00%, which is second place. The author also classified the x-ray images with many CNN architectures, such as DenseNet201,ResNet50, EfficientNetB0, InceptionResNetV2, InceptionV3, ResNet18, ShuffleNet, and SqueezeNet. However, the ResNet50 showed the best result with only 98.99% accuracy.

However, the author should be more concerned about these comments.

  • In the title, please remove '(BoDMCF).'
  • In our approach section, it is quite a long paragraph. Please split it.
  • In the novelty section, please clarify why the proposed method is a novel method. Because many papers also extracted features using the CNN model.
  • Could the author provide the section of related work? Moreover, provide more detail or use a subsection such as 3. Related work, 3.1 CNN for breast cancer tumor classification 3.2 Extract deep features from breast cancer tumor images, etc.
  • As shown in Table 11, what is the 'Efficientnetbo $ Inception-V3?
  • Please check the consistency of the name of the CNN architecture, such as InceptionV3 or Inception-V3, ResNet-50 or Resnet50, etc.
  • As described in Section 3.1, 
    • "the brightness of the image randomly between 50% (1-0.5) and 150% (1+0.5) of the original image." Could the author show example of the images after applying the brightness technique?
    •  "Data augmentation generates diverse samples, which enables the model to learn a robust representation that is invariant to minor changes." Which data augmentation techniques were used in this manuscript?
  • As shown in Figure 6, in the "Deep CNN Feature Extractors", Does the author extract the features using three CNN architectures at the same time? If the author did not use three CNN architectures at the same time, the author should not have a specific architecture in the "Deep CNN Feature Extractors" block.
  • The author presented using "the bag of deep multiresolution CNN features" However, lack of information on how does the author create a bag of deep features? Or the author used only the pooling method, not a bag of feature method. Please clarify it.
  • The author experiments on the BreakHis breast cancer image dataset containing 7909 images of benign and malignant breast tumors. The dataset was split into training and test sets with a ratio of 70:30. Does the author concerned about the significance of the result? So, why did the author not use the cross-validation method? The author will see the average result and standard deviation with the cross-validation method and please check the significance value of the proposed method.
  • The author presented a confusion matrix (see Figure 12). Only two images were misclassified. Could the author show the misclassified images and discuss why it is misclassified? It is interesting to discuss it.
  • The author also experimented with the resolution number, such as 40X, 100X, 200X, and 400X. The results showed that 100X outperformed other resolutions. Could the author discuss this issue?
  • Figure 9, could the author remove "date"? Also, the caption of Figure 9 is "Efficientnetbo Train-Test performance." However, the title of the graph is "Training Progress." Please correct it.

Author Response

Dear reviewer,

Thank you so much for reviewing our manuscript and for your thoughtful comments. We have carefully reviewed, thought about the reviewers' comments and responded to them below. In this rebuttal document, our responses in black are clarifications. Our responses in red indicate that we made some changes to the manuscript to address the corresponding comment. An updated version of our paper is also attached. In the manuscript, text in red is either new text we inserted to effect the changes outlined in this rebuttal (in red) or text we already had in the    previous submission, but we are highlighting as it addresses some of the reviewers’ comments.

We are extremely grateful for the time you spent reviewing this manuscript once again, and your  well-thought-out comments. We believe your comments improved the paper significantly!

Please see attachment for our point-by-point response to your comments

Reviewer 2 Report

This paper proposed an integrated method for Tumor Classification using Multi-Resolution CNNs features were extracted from histopathological images. It could be useful to researchers in CAD system, and in general to researchers working in such area.  I have to state that I have a positive opinion about this study in consequence of its importance. However, I would like to draw the attention of the authors to the important points that need to be corrected in the article. Before acceptance, the following points must be incorporated.

1.     Abstract must become more concise. In the abstract, it is necessary to emphasise the contributions of the research relative to existing studies in the field.

2.     Why they mentioned results in “our approach” of the introduction.

3.     Author(s) should explain more clearly about four resolutions (40X, 100X, 200X and 400X) and how they very to each other compare to original ones.

4.     Figure 9, its make me confuse. Which colour is for “train” and/or “test”? Explanation required…

5.     Did you use any validation dataset for model validations before feature extraction and during SVM implementations?

6.     In Figure 6, multi-model features were downsized using pooling. Then features were fed into SVM. It may have overfit the SVM training. I’m very much interested in knowing how they overcame the overfitting challenge. In particular, how were the values C and gamma chosen?

7.     It is more important to describe how their technique can be implemented in a real-time applications.

8.     In addition, is the increase of evaluations through the independent and their integration are statistical significant?

9.     Why did they use SVM when CNN is the state-of-the-art in AI applications?

10. Conclusion must more precise

@ In the related study section, the author should add recent articles which have merged CNN outputs with traditional machine learning. I’ll suggest adding the following related articles with proper descriptions:

1. “Digital mammographic tumor classification using transfer learning from deep convolutional neural networks”

2. “Detection and Visualisation of Pneumoconiosis Using an Ensemble of Multi-Dimensional Deep Features Learned from Chest X-rays”

3. Ensemble learners of multiple deep CNNs for pulmonary nodules classification using CT images

4. ‘Computer-Aided Diagnosis of Coal Workers’ Pneumoconiosis in Chest X-ray Radiographs Using Machine Learning: A Systematic Literature Review’

5. “Use of clinical MRI maximum intensity projections for improved breast lesion classification with deep convolutional neural networks”

6. “Automated detection of pneumoconiosis with multilevel deep features learned from chest X-Ray radiographs”

7. “Leukemia diagnosis in blood slides using transfer learning in CNNs and SVM for classification”

Author Response

(The authors gave the same response as above.)

Reviewer 3 Report

The paper proposed a hybrid deep learning based method for breast cancer recognition. The results of experiments on a benchmark dataset of histopathological images are presented. The technical quality and presentation should be improved before the paper could be considered for publication.

Comments:

1.      Improve the abstract: add the percentage sign to the numbers expressed in percentage units.

2.      Some known deep learning models (ResNet-50, InceptionV3 and Efficientnet-b0) for feature extraction and a widely used SVM for classification, which also have been used before with various modifications and optimizations. Clearly state your novelty and difference from previous works at the end of the Introduction section.

3.      Improve the literature review. The structure is poor. The selection of works seems to be ad hoc. Some of the works discussed are outdated. See a survey of methods and works on breast cancer recognition in Systematic review of computing approaches for breast cancer detection based computer aided diagnosis using mammogram images. For specific methods, see TTCNN: A breast cancer detection and classification towards computer-aided diagnosis using digital mammography in early stages. Medical internet-of-things based breast cancer diagnosis using hyperparameter-optimized neural networks. Complete the section by a discussion on the limitations of the previous works.

4.      Did you use any image preprocessing such as denoising or histogram enhancement? Noisy images are a well-known problem in biomedical imaging which must be addressed.

5.      A very good result shown in Table 13 may be indicative of overfitting. How do you avoid/prevent overfitting during training?

6.      Fig. 9 is not much informative. I suggest to remove,

7.      Is the difference between model performance presented in Tables 7-13 statistically significant? Present the statistical analysis of the results. To rank models, you can use the post-hoc Nemenyi test and visualize using the Critical Distance diagram.

8.      Consider explainability of the results using for example Grad-CAM and discuss.

9.      Discuss cases of misclassifications (with figures) visible from Fig. 12.

10.  Discuss the limitations of your method.

11.  Improve the conclusions section. Currently, it only summarizes the work done. Provide deeper insights into the implications of this study for the biomedical imaging research field.

Author Response

Dear reviewer,

Thank you so much for reviewing our manuscript and for your thoughtful comments. We have carefully reviewed, thought about the reviewers' comments and responded to them below. In this rebuttal document, our responses in black are clarifications. Our responses in red indicate that we made some changes to the manuscript to address the corresponding comment. An updated version of our paper is also attached. In the manuscript, text in red is either new text we inserted to effect the changes outlined in this rebuttal (in red) or text we already had in the    previous submission, but we are highlighting as it addresses some of the reviewers’ comments.

We are extremely grateful for the time you spent reviewing this manuscript once again, and your   well-thought-out comments. We believe your comments improved the paper significantly!

Please see attachment for our point-by-point response to your comments.

Round 2

Reviewer 1 Report

The revised manuscript is improved and reached the quality to publish in the journal. I am very satisfied with this version.

- Line 97, please remove 'Our work is novel because.'

Author Response

Dear editor,

Thank you so much for reviewing our manuscript and for your thoughtful comments. We have carefully reviewed, thought about the reviewers' comments and responded to them below. In this rebuttal document, our responses in black are clarifications. Our responses in red indicate that we made some changes to the manuscript to address the corresponding comment. An updated version of our paper is also attached. In the manuscript, text in red is either new text we inserted to effect the changes outlined in this rebuttal (in red) or text we already had in the submission, but we are highlighting as it addresses some of the reviewers’ comments.

We are extremely grateful for the time you spent reviewing this manuscript once again, and your well-thought-out comments. We believe your comments improved the paper significantly!

Reviewer 2 Report

The author(s) revised effectively, and I am pleased with their response. Minor corrections are:

1. Table 9 should be before 10.

2. Do not need to revel your findings in the introduction section. The reader will probably be most interested in the sections on the results, discussion, and conclusion  for this.

Author Response

Dear reviewer,

Thank you so much for reviewing our manuscript and for your thoughtful comments. We have carefully reviewed, thought about the reviewers' comments and responded to them below. In this rebuttal document, our responses in black are clarifications. Our responses in red indicate that we made some changes to the manuscript to address the corresponding comment. An updated version of our paper is also attached. In the manuscript, text in red is either new text we inserted to effect the changes outlined in this rebuttal (in red) or text we already had in the submission, but we are highlighting as it addresses some of the reviewers’ comments.

We are extremely grateful for the time you spent reviewing this manuscript once again, and your well-thought-out comments. We believe your comments improved the paper significantly!

Reviewer 3 Report

The authors have revised well. I have only a few technical comments:

- Check Equation (16) for correctness.

- Table 7, Table 8: emphasize (bolden) the best results.

- Equations 23, 24: explain notations and variables.

Author Response

(The authors gave the same response as above.)
